# Study on Battery Charging Converter for MPPT Control of Laser Wireless Power Transmission System

**Seongjun Lee** [1]**, Namgyu Lim** [2]**, Wonseon Choi** [3]**, Yongtak Lee** [3]**, Jongbok Baek** [4] 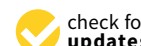 **and Jungsoo Park** [1,*]

1. Department of Mechanical Engineering, Chosun University, Gwangju 61452, Korea; lsj@chosun.ac.kr
2. Department of Mechanical System & Automotive Engineering, Chosun University, Gwangju 61452, Korea; lnk9100@chosun.kr
3. Gwangju Institute of Science and Technology, Gwangju 61452, Korea; bourne.ws.choi@gmail.com (W.C.); ytlee@gist.ac.kr (Y.L.)
4. Energy ICT Convergence Research Department, Korea Institute of Energy Research, Daejeon 34129, Korea; jongbok.baek@kier.re.kr
* Correspondence: j.park@chosun.ac.kr; Tel.: +82-62-230-7057

**Abstract:** Herein, the voltage and current output characteristics of a laser photovoltaic (PV) module applied to a wireless power transmission system using a laser beam are analyzed. First, an experiment is conducted to obtain the characteristic data of the voltage and current based on the laser output power of the laser PV module, which generates the maximum power from the laser beam at a wavelength of 1080 nm; subsequently, the small-signal voltage and current characteristics of the laser PV module are analyzed. From the analysis results, it is confirmed that the laser PV module has a characteristic in which the maximum power generation point varies according to the power level of the laser beam. In addition, similar to the solar cell module, it is confirmed that the laser PV module has a current source and a voltage source region, and it shows a small signal resistance characteristic having a negative value as the operating point goes to the current source region. In addition, in this paper, by reflecting these electrical characteristics, a method for designing the controller of a power converter capable of charging a battery while generating maximum power from a PV module is proposed. Since the laser PV module corresponds to the input source of the boost converter used as the power conversion unit, the small-signal transfer function of the boost converter, including the PV module, is derived for the controller design. Therefore, by designing a controller that can stably control the voltage of the PV module in the current source, the maximum power point, and voltage source regions defined according to the output characteristics of the laser PV module, the maximum power is generated from the PV module. Herein, a systematic controller design method for a boost converter for laser wireless power transmission is presented, and the proposed method is validated based on the simulation and experimental results of a 25-W-class boost converter based on a microcontroller unit control.

**Keywords:** laser wireless power transmission; PV module; maximum power point; battery charging

## 1. Introduction

Recently, studies regarding wireless charging technology for supplying electric power to electric vehicles, various IoT (Internet of Things) devices, and unmanned moving objects have been actively conducted. Hitherto, wireless charging technology has been mainly studied based on magnetic induction, magnetic resonance, and electromagnetic wave methods. The magnetic induction method is mainly used when the distance between the transmitting and receiving coils is short (1–2 cm for

electronic products or 0.15 m for electric vehicles), and the magnetic resonance method is applied when the transmission distance is longer than the magnetic induction method. In this case, the main target is a distance of less than 10 m. The electromagnetic wave method enables power to be transmitted up to a transmission distance of several kilometers compared with the other two methods; however, its transmission efficiency is low, and it is harmful to the human body [1–4].

On the other hand, the wireless charging method using the laser is also being studied in military and space applications [2,5]. Figure 1 shows a schematic diagram of an unmanned aerial vehicle (UAV) wireless power transmission system using a laser beam proposed in [5]. The light energy of the laser beam irradiated from the ground is transferred to the laser photovoltaic (PV) module installed inside the UAV and converted into electric energy. The converted electrical energy is used to charge the battery through a power converter or as energy for components mounted on the UAV. Research on the laser wireless charging system to date is as follows.

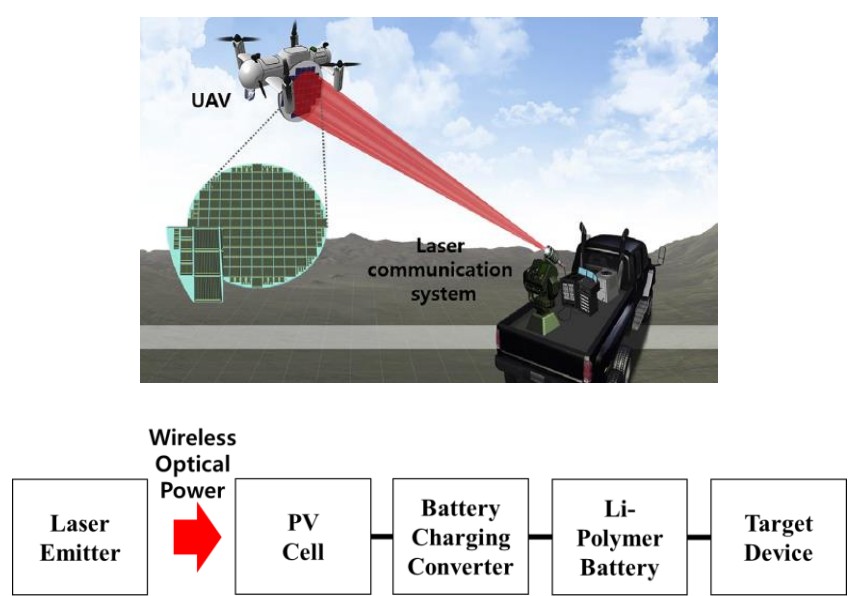

**Figure 1.** Laser wireless charging system configuration of unmanned aerial vehicle (UAV) [5].

Results pertaining to the voltage, current, and efficiency based on the material type used and the laser wavelength in a laser PV cell are presented in [5,6]. However, in addition to these results, a method to control a converter that can charge a battery when a laser PV module is applied to a UAV is necessitated. Results pertaining to the wavelength and temperature output characteristics of PV panels applied to charging systems using laser beams as well as the mathematical modeling of PV panels are presented in [7]. In this study, the modeling of the laser PV module proposed in [7] was used to design a PV module applied to a simulation model; however, the output characteristics were analyzed through testing a prototype laser PV module. In addition, a control design method that can generate the maximum power from a laser PV module using a battery charging converter was investigated. In [8], the concept of the maximum power point tracking (MPPT) algorithm applied with a neural network (NN) algorithm considering the characteristics of a PV module, and a laser beam was presented. However, in that paper, the improvement compared with perturb-and-observe and incremental conductance methods, as well as the specific design method and experimental results of NN, were not presented.

Similar to the previous studies mentioned above, studies regarding laser wireless charging systems mainly focus on improving the efficiency based on the material of the laser PV module and the introduction of research areas where the laser charging system is applicable. However, for the practical use of the laser charging system, studies regarding the control of the battery charging system using laser PV modules is required. There has been a lot of research on battery charging systems using

solar cell modules, but there are no research articles on the power converter design and control of laser wireless charging systems using laser beams. Therefore, this paper deals with a controller design method based on the experimental results of laser PV module output characteristics for controlling the power converter of a laser wireless charging system.

To extract the maximum power from a laser PV module, the operating point of the PV module must be controlled. To design a controller for controlling the operating point of the input source, a small-signal model of the PV module is required in the current source, voltage source, and maximum power point regions based on the laser output. First, we conducted an experiment to investigate voltage and current characteristics based on the laser output power of a laser PV module to generate the maximum power from a laser of a specific wavelength; subsequently, we derived the resistance values for small-signal fluctuations of voltage and current. After deriving the transfer function of the boost converter, including the small-signal resistance of the laser PV module, we designed a controller that can satisfy the dynamic performance in the entire operational area of the laser PV module. As described above, we herein present a method for designing the small-signal transfer function of a boost converter that reflects the small-signal characteristics of a laser PV module and a controller design method that can generate the maximum power from a PV module based on the MPPT algorithm. The method proposed herein was validated based on the simulation and experimental results of a 25-W-class boost converter prototype.

## 2. Modeling and Controller Design of Laser Wireless Power Transmission System

### 2.1. Laser PV Module Characteristics

The PV module, which is the input source of the laser wireless charging system, has a configuration in which 16 PV cells are connected in series, as shown in Figure 2; it was manufactured by the Korea Advanced Nano Fabrication Center [5]. Using a high-power continuous fiber laser MFSC-200, laser beams with a wavelength of 1080 nm were irradiated onto the laser PV module at 2.478 and 2.874 W/cm$^2$ [9]. Figure 3 shows the voltage–current and voltage–power characteristics based on the laser beam power used in the experiment.

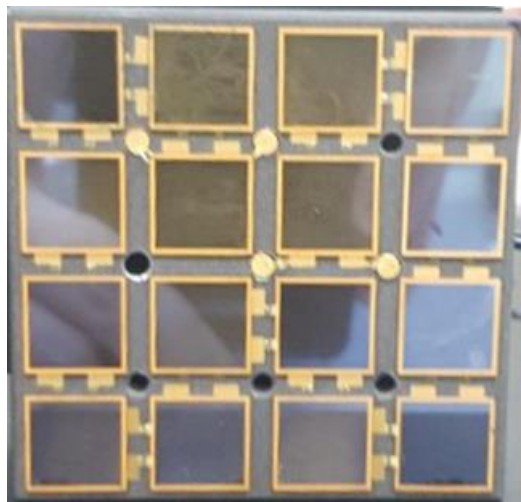

**Figure 2.** Laser photovoltaic (PV) module with 16 cells connected in series.

As shown in Figure 3, the PV module presents nonlinear output characteristics, in which the short-circuit current, open-circuit voltage, and location of the maximum power generation point varied according to the laser beam power. Since the maximum power generated from the PV module is the point at which the slope becomes 0 in the voltage–power curve, it can be confirmed that the maximum power generation point measured in the experiment is approximately 7 V. To quickly charge the UAV's

battery from the PV module, the battery charging converter must include an MPPT algorithm to identify the maximum power point of the laser PV module as well as stably control the operating point of the input source determined by the algorithm. Therefore, the small-signal characteristics of the laser PV module must be analyzed to control the operating point of the input source; the small-signal resistance (denoted as $r_s$) for the increment of the measured voltage and current of the laser PV module is shown in Figure 4. As shown, the laser PV module exhibits negative small-signal resistance characteristics, and the small-signal resistance value increases as it approaches the current source region. In the next section, a method for deriving the small-signal transfer function of a boost converter that reflects the small-signal resistance characteristics of a laser PV module and a controller design method are presented.

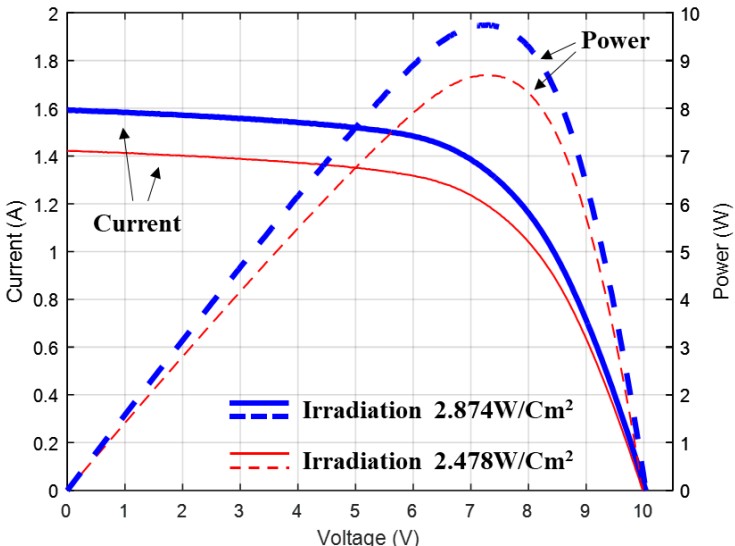

**Figure 3.** Voltage–current and voltage–power characteristics based on laser power of PV module.

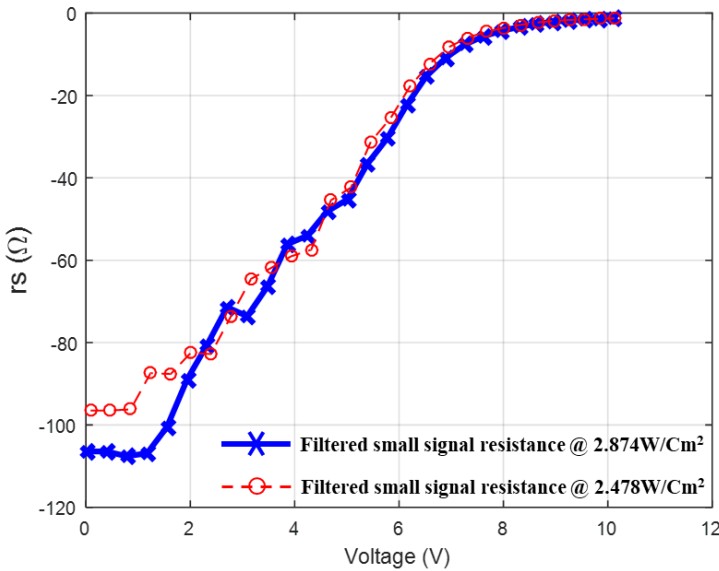

**Figure 4.** Small-signal resistance characteristics of laser PV module.

### 2.2. Modeling and Controller Design of Boost Converter

Figure 5 shows the boost converter topology of the battery charging system to which the laser PV module was applied. In Figure 5, L is the inductor of the boost converter, C is the input capacitor, $r_c$ is the capacitor's equivalent series resistance (ESR), and $V_b$ is the voltage source representing the

battery connected to the output. The boost converter must have an operating point control function to generate the maximum power from the PV module before the battery reaches its full voltage [10–12]. In this study, the boost converter was set to control the voltage of the PV module such that the laser PV module can form a stable operating point in a large signal. For the controller design and stability analysis of the boost converter, a small-signal modeling technique using the state-space averaging method was applied, which is a method of obtaining the transfer function from the input to the output of the system by linearizing the system based on the operating point in the steady state [13,14].

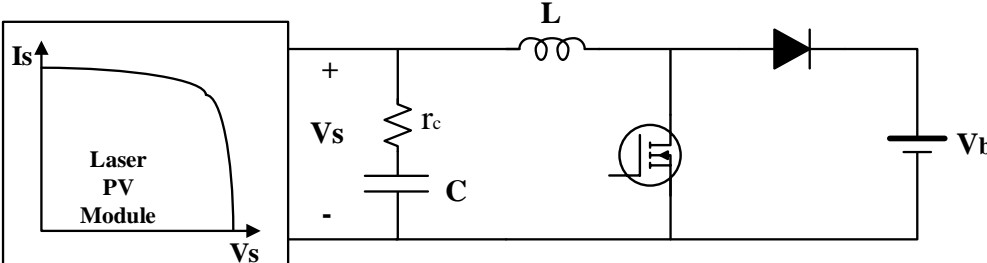

**Figure 5.** Boost converter for battery charging with laser PV module.

The small-signal transfer function required for designing the voltage controller of the laser PV module of the boost converter was derived in the following two steps. First, as shown in Figure 6, a small-signal model of the unterminated model for the boost converter modeling of the PV module as a current source was developed. Subsequently, the small-signal transfer function of the boost converter to which the laser PV module was applied was derived by including the relationship between the small-signal voltage and current of the laser PV module mentioned in Section 2.1.

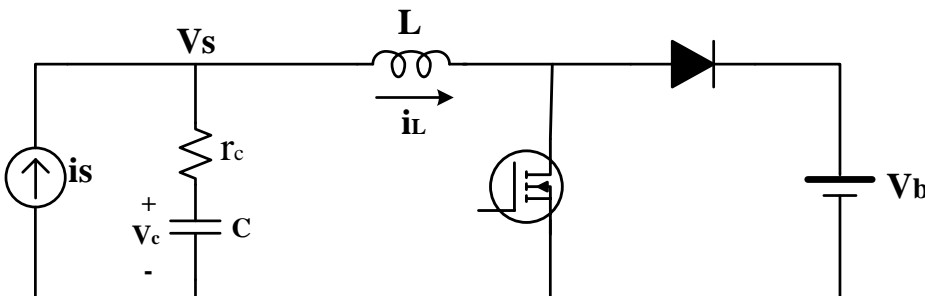

**Figure 6.** Unterminated model of boost converter with input as the current source.

The state equation obtained by averaging the differential equations of the voltage and current based on the switching states of the power switch of the boost converter in Figure 6 is shown in Equation (1), where $d$ denotes the ratio of turning ON during one period of the switching frequency, and $d'$ is defined as $1 - d$.

$$\begin{bmatrix} \dot{i}_L \\ \dot{v}_c \end{bmatrix} = \begin{bmatrix} -\frac{r_c}{L} & \frac{1}{L} \\ -\frac{1}{C} & 0 \end{bmatrix} \begin{bmatrix} i_L \\ v_c \end{bmatrix} + \begin{bmatrix} \frac{r_c}{L} & -\frac{1}{L}d' \\ \frac{1}{C} & 0 \end{bmatrix} \begin{bmatrix} i_s \\ v_b \end{bmatrix} \tag{1}$$

To develop the small-signal model, the state and control variables of the boost converter were defined as having small perturbations at the steady-state operating point, which is indicated by capital letters as follows: $v_c = V_c + \hat{v}_c$, $i_L = I_L + \hat{i}_L$, $d = D + \hat{d}$, $v_b = V_b + \hat{v}_b$ and $i_s = I_s + \hat{i}_s$. Since the small perturbation excluding the steady-state operating point yields a small-signal model, the small-signal

transfer function of the boost converter is as shown in Equation (2). In addition, the small-signal block diagram of the output to the input can be represented as shown in Figure 7.

$$
\begin{cases}
G_1 = \dfrac{\hat{i}_L}{\hat{i}_s} = \dfrac{(1+sr_cC)}{\Delta} \\[2mm]
G_2 = \dfrac{\hat{i}_L}{\hat{v}_b} = -\dfrac{sCD'}{\Delta} \\[2mm]
G_3 = \dfrac{\hat{v}_s}{\hat{i}_s} = \dfrac{sL(1+sr_cC)}{\Delta} \\[2mm]
G_4 = \dfrac{\hat{v}_s}{\hat{v}_b} = D'\dfrac{(1+sr_cC)}{\Delta} \\[2mm]
G_5 = \dfrac{\hat{i}_L}{\hat{d}} = \dfrac{sCV_b}{\Delta} \\[2mm]
G_6 = \dfrac{\hat{v}_s}{\hat{d}} = -V_b\dfrac{(1+sr_cC)}{\Delta}
\end{cases}
\tag{2}
$$

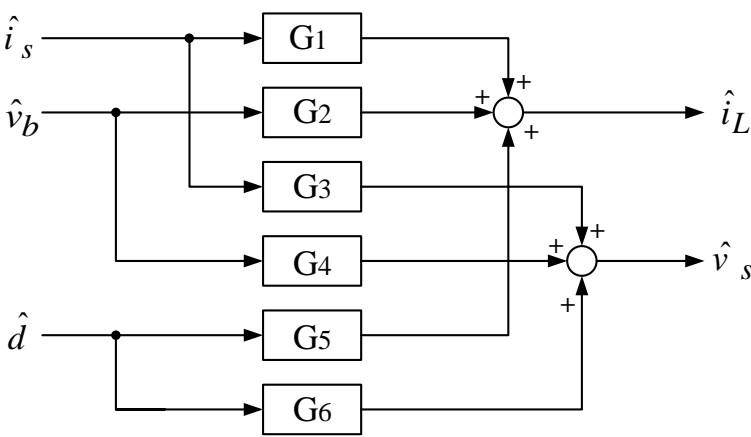

**Figure 7.** Small-signal block diagram of unterminated model.

The characteristic equation $\Delta$, Q-factor, and resonance frequency of the transfer function are expressed as shown in Equation (3).

$$
\begin{cases}
\Delta = 1 + \dfrac{s}{Q\omega_o} + \dfrac{s^2}{\omega_o^2} \\[2mm]
Q = \dfrac{1}{r_c}\sqrt{\dfrac{L}{C}} \\[2mm]
\omega_o = \dfrac{1}{\sqrt{LC}}
\end{cases}
\tag{3}
$$

As mentioned in Section 2, the laser PV module represents nonlinear voltage and current characteristics; therefore, a small-signal model can be obtained through the Taylor series for the operating point, as shown in Equation (4).

$$
\begin{cases}
i_s = f(v_s) \\[1mm]
I_s + \hat{i}_s = f(V_s + \hat{v}_s) = f(V_s) + \dot{f}(V_s)\cdot\hat{v}_s \\[1mm]
\hat{i}_s = r_s\hat{v}_s \quad (r_s < 0)
\end{cases}
\tag{4}
$$

In the unterminated model, the relationship between $\hat{v}_s$ and $\hat{i}_s$ corresponds to the small-signal resistance $r_s$ of the laser PV module; therefore, the small-signal block diagram in Figure 7 can be reorganized as shown in Figure 8. As depicted in Figure 8, when the laser PV module is connected to the input of the boost converter, a loop gain T is formed where $\hat{v}_s$ is fed back to $\hat{i}_s$; therefore, the final

small-signal transfer function of the boost converter considering the laser PV module can be derived from Equation (5).

$$
\begin{cases}
\frac{\hat{v}_s}{\hat{v}_b} = \frac{G_4}{1+T} \\
\frac{\hat{v}_s}{\hat{d}} = \frac{G_6}{1+T} \\
\frac{\hat{i}_L}{\hat{v}_b} = G_2 + \frac{G_1}{r_s} \frac{G_4}{1+T} \\
\frac{\hat{i}_L}{\hat{d}} = G_5 + \frac{G_1}{r_s} \frac{G_6}{1+T}
\end{cases}
\tag{5}
$$

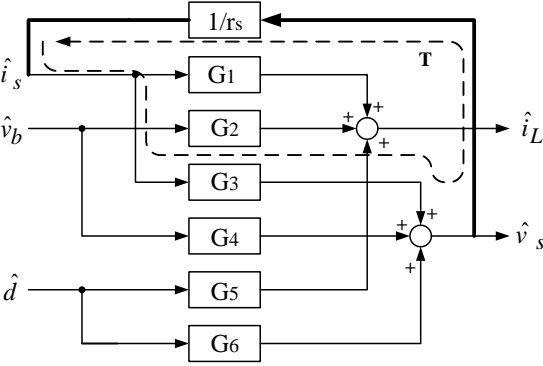

**Figure 8.** Small-signal block diagram with added PV module.

Therefore, the final transfer function of the boost converter considering the PV module is expressed as shown in Equations (6)–(9), and the small-signal block diagram including the inductor current of the boost converter and the input voltage controller of the laser PV module is as shown in Figure 9. To control the input voltage with the voltage command generated by the MPPT algorithm that tracks the maximum power point of the laser PV module, a two-loop controller structure was adopted, in which a voltage controller was applied to the outer loop while an internal current controller was applied. Since the converter is operated by a digital controller through the micro controller unit (MCU), the transfer function of the block diagram is expressed as a discrete-time transfer function.

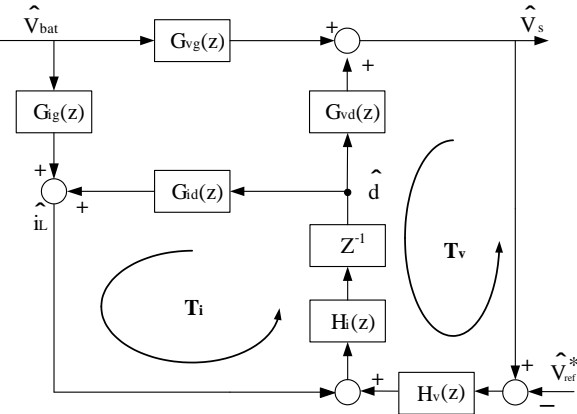

**Figure 9.** Small-signal block diagram for two-loop control.

The open-loop current gain (Ti) for controlling the inductor current of the boost converter is shown in Figure 9. At this time, for digital control, the continuous-time small-signal transfer function obtained previously was converted into a discrete-time transfer function using the zero-order hold (ZOH) method, and the conversion equation is as shown in Equation (10). Here, $Z\{\cdot\}$ represents the

z-transform, $(1 - e^{-sT_s})/s$ represents the transfer function of the ZOH, and $Gp(s)$ represents the transfer function of Equations (6)–(9).

$$\frac{\hat{v}_s}{\hat{v}_b} = G_{vg} = D' \frac{(1 + sr_cC)}{1 + s\left(r_cC - \frac{L}{r_s}\right) + s^2\left(LC - \frac{r_cLC}{r_s}\right)} \tag{6}$$

$$\frac{\hat{i}_L}{\hat{v}_b} = G_{ig} = \frac{D'}{r_s} \frac{(1 - sr_sC)}{1 + s\left(r_cC - \frac{L}{r_s}\right) + s^2\left(LC - \frac{r_cLC}{r_s}\right)} \tag{7}$$

$$\frac{\hat{v}_s}{\hat{d}} = G_{vd} = -V_b \frac{(1 + sr_cC)}{1 + s\left(r_cC - \frac{L}{r_s}\right) + s^2\left(LC - \frac{r_cLC}{r_s}\right)} \tag{8}$$

$$\frac{\hat{i}_L}{\hat{d}} = G_{id} = -\frac{V_b}{r_s} \frac{(1 - sr_sC)}{1 + s\left(r_cC - \frac{L}{r_s}\right) + s^2\left(LC - \frac{r_cLC}{r_s}\right)} \tag{9}$$

$$G_p(z) = Z\left\{\frac{\left(1 - e^{-sT_s}\right)}{s}G_p(s)\right\} \tag{10}$$

In this study, a proportional and integral (PI) controller was used for current control, as shown in Equation (11), and the Ti applied with the PI current controller is shown in Equation (12). At this time, $z^{-1}$ represents the time delay until the calculated duty is reflected, and it was considered as a one-sampling delay in this study. Figure 10 shows the Bode diagram of the Ti of the designed current controller. CSR denotes the current-source region of the laser PV module, MPP denotes the region near the maximum power point, and VSR denotes the voltage-source region. Since the laser PV module has nonlinear characteristics, it must be designed to ensure stability in the CSR, VSR, and MPP of the laser PV module, as shown in the Bode diagram.

$$H_i(z) = 0.075\frac{(z - 0.93)}{(z - 1)} \tag{11}$$

$$T_i = G_{id}(z) \cdot H_i(z) \cdot z^{-1} \tag{12}$$

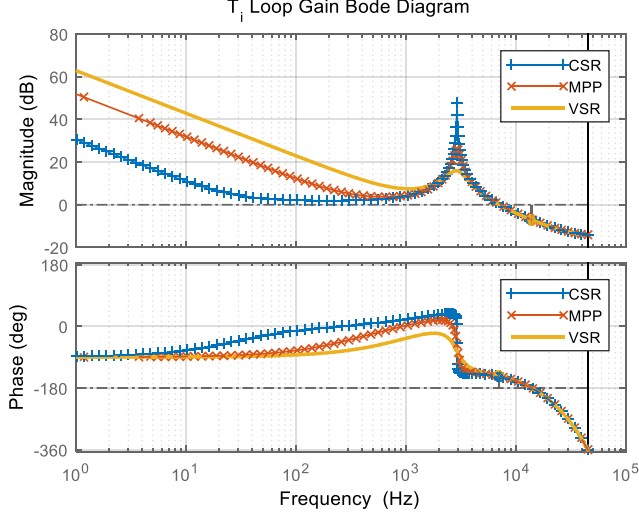

**Figure 10.** Bode diagram of open-loop current gain.

The voltage controller of the laser PV module was designed to achieve dynamic performances (bandwidth) and stability (phase margin) for a system with a closed current loop, as shown in Equation (13). The designed PI voltage controller is expressed as shown in Equation (14); Equation (15) shows

the equation of the open-loop voltage gain when the designed voltage controller is applied. As shown from the voltage loop gain of Figure 11, even when the operating point of the laser PV module has changed, the voltage controller maintained a bandwidth of 400–700 Hz, and the phase margin is designed to exceed 70°.

$$\frac{V_{sa}}{V_c} = G_{vdc}(z) = \frac{G_{vd}(z){\cdot}H_i(z){\cdot}z^{-1}}{1 + T_i} \tag{13}$$

$$H_v(z) = 0.75\frac{(z - 0.979)}{(z - 1)} \tag{14}$$

$$T_v = G_{vdc}(z){\cdot}H_v(z) \tag{15}$$

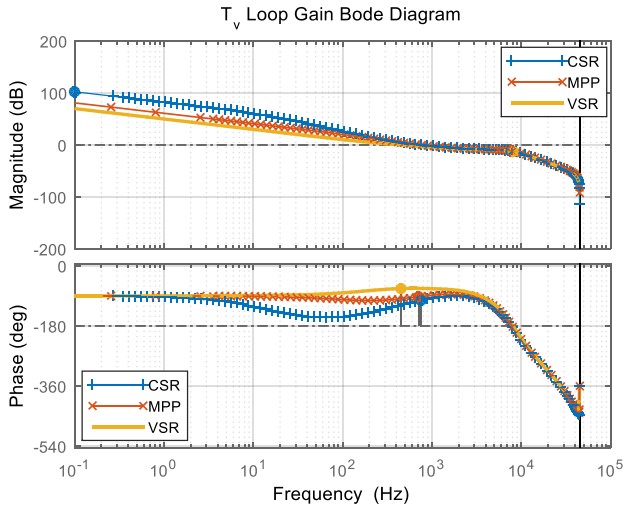

**Figure 11.** Board diagram of voltage loop gain of two-loop control.

## 2.3. MPPT Algorithm Design

In this study, the incremental conductance method with variable amplitude was applied to track the maximum power point based on the output power of the laser light source [15]. This method compares the load impedance with the impedance of the laser PV module and controls the voltage of the PV module to correspond to the maximum power point. As shown in Figure 12, when the output of the PV module is located to the left of the maximum power point, the power increases with the voltage. Conversely, when it is located to the right of the maximum power point, the power decreases with the increase in the voltage. This relationship is expressed as shown in Equations (16)–(19).

Therefore, the voltage command of the laser PV module can be positioned as the maximum power point by measuring and comparing the increment for the conductance of the laser PV module and the instantaneous resistance value. At this time, the fluctuation value of the command value was set as a variable voltage such that when the operating point was far from the maximum power point, it rapidly converged to the maximum power point, and in the vicinity of the maximum power point, the periodic vibration width reduced compared with using a fixed value. As described above, Figure 13 shows the flow chart of the MPPT algorithm with variable amplitude based on the incremental conductance method.

$$\frac{dP}{dV} = \frac{d(IV)}{dV} = I + V\frac{dI}{dV} \cong I + V\frac{\Delta I}{\Delta V} \tag{16}$$

$$\frac{\Delta I}{\Delta V} = -\frac{I}{V} \quad at\ MPP \tag{17}$$

$$\frac{\Delta I}{\Delta V} > -\frac{I}{V} \quad left\ of\ MPP \tag{18}$$

$$\frac{\Delta I}{\Delta V} < -\frac{I}{V} \quad right\ of\ MPP \tag{19}$$

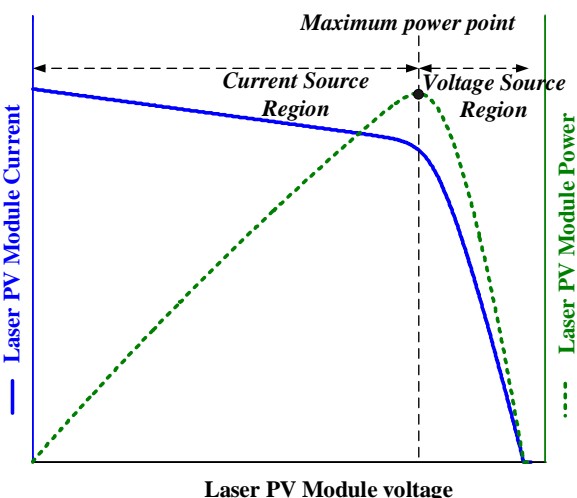

**Figure 12.** Characteristic curve of the laser PV module.

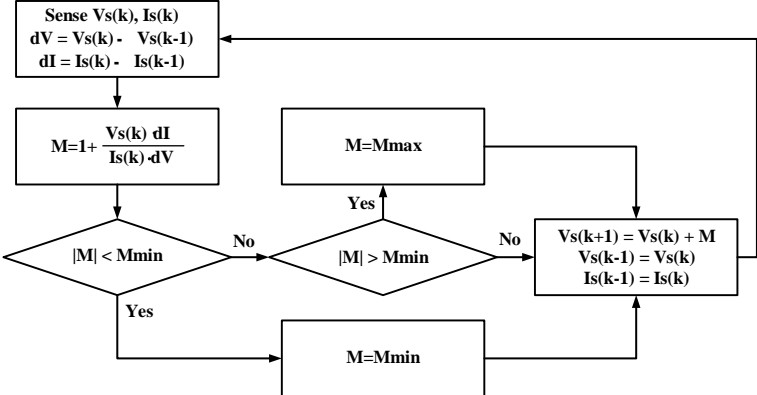

**Figure 13.** Algorithm flow chart of an incremental conductance method with variable amplitude.

## 3. Simulation and Experimental Results

### 3.1. Simulation Result

The method proposed herein was first verified using the simulation model shown in Figure 14. In the simulation model, the laser module was modeled with a diode model-based equation [7,16]. The laser PV module output current can be described as shown in Equations (20)–(22) in Figure 15a, where $I_{sc}$ is the short circuit current of the laser PV module, $I_o$ is the reverse saturation current of the diode, $q$ is the electronic charge, $v_d$ is the diode voltage, $K$ is the Boltzmann constant, $T$ is the temperature in Kelvin, $n$ is the ideality factor, $R_{sh}$ is the shunt resistance, $R_s$ is the series resistance, $v_{oc}$ is the open circuit voltage of the cell, and $Ns$ is the serial number of cells constituting the module. Figure 15b shows the modeling of the laser PV module implemented in Matlab/Simulink using Equations (20)–(22). Illumination is a variable representing the laser power intensity irradiated to the laser PV module and has a value of 0 to 1 normalized to 2.874 W/cm$^2$.

$$I_s = I_{sc} - I_o\left(e^{\frac{qv_d}{nKT}} - 1\right) - \frac{v_d}{R_{sh}} \tag{20}$$

$$I_o = \left(I_{sc} - \frac{v_{oc}}{R_{sh}}\right)\bigg/\left(e^{\frac{qv_{oc}}{nKT}} - 1\right) \tag{21}$$

$$v_s = N_s(v_d - R_sI_s) \tag{22}$$

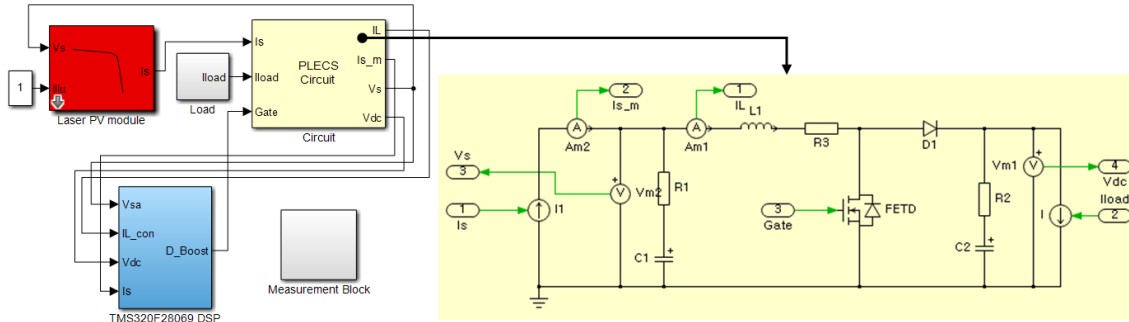

**Figure 14.** Simulation model using Matlab/Simulink.

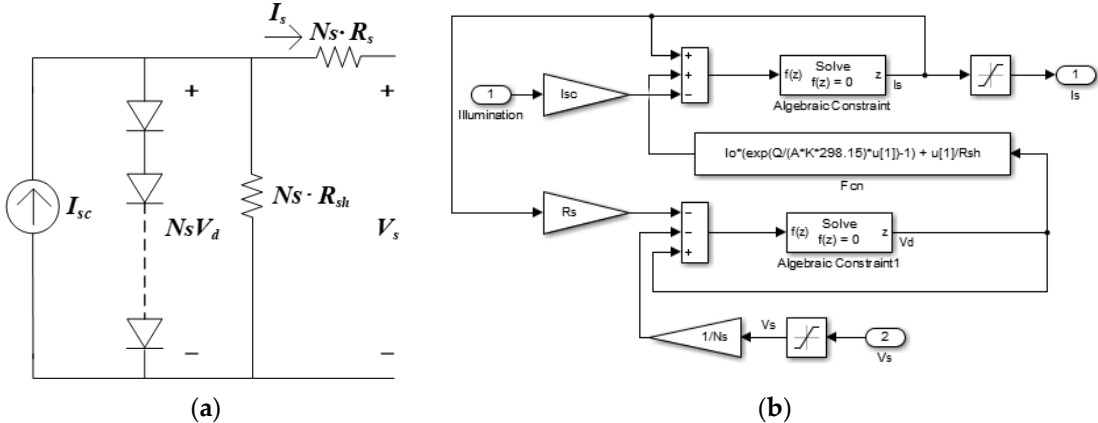

(**a**)                                                                 (**b**)

**Figure 15.** Modeling of laser PV module: (**a**) equivalent circuit model (**b**) simulation model in Matlab.

The parameters used for modeling the laser PV module are shown in Table 1. In the parameters of the laser PV module, $I_{sc}$ and $V_{oc}$ were set as the short-circuit current and open-circuit voltage values measured in the experiment, and the $R_s$, $R_{sh}$, and $n$ values were extracted by the trial and error method through the simulation model in Figure 15b. In order to extract more accurate parameters of the laser PV module using an optimization algorithm, the method mentioned in Sheng and Anani's articles can be applied [17,18]. The accuracy of the simulation model compared with the experimental data of the module mentioned in Section 2 is shown in Figure 16. As can be seen in Figure 16a, the root mean square error (RMSE) of the simulation and experimental results is shown as 0.0027 or less, and the simulation model reflects the experimental results well.

**Table 1.** Parameters of laser PV module in simulation model.

| Parameters | Values | Parameters | Values |
|---|---|---|---|
| $I_{sc}$ | 1.61 A @ 2.478 W/cm$^2$ | $q$ | $1.602 \times 10^{-19}$ C |
| $V_{oc}$ | 0.628 V/cell | $K$ | $1.381 \times 10^{-23}$ JK$^{-1}$ |
| $R_s$ | 50 mΩ/cell | $T$ | 298.15 K |
| $R_{sh}$ | 4.5 Ω | n | 1.8 |
| $Ns$ | 16 | | |

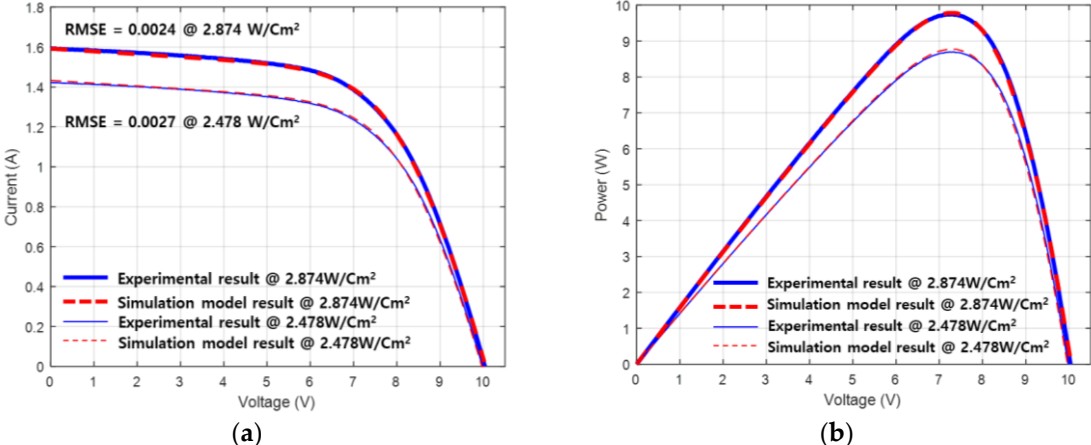

**Figure 16.** Characteristics of laser PV module using simulation model: (**a**) V–I characteristics comparison between the simulation model and experimental results; (**b**) V–P characteristics of the simulation model.

The power circuit of the boost converter for charging a 24 V battery was modeled using PLECS software [19] and implemented in the PLECS circuit block, as shown in Figure 14. In the boost converter, the battery was modeled with a capacitor and an ESR, and the value was set to 2 F, which afforded a smaller capacity than the actual battery to reduce the simulation time. The parameter values of the components applied to the simulation are summarized in Table 2.

**Table 2.** Experimental setup of the laser charging system.

|  | **Name of Device/Manufacturer** | **Rating/Values** |
|---|---|---|
| Laser Source | Max MFSC 200 W-300 L Air Cooling Fiber | Power range 0–140 W<br>Central wavelength 1080 nm |
| Laser Receiving Panel | Custom-made PV module | Voc: 10 V, Isc: 1.6 A @ 2.478 W/cm$^2$ |
| Electronic Load | ITECH LT8511 | 120 V/30 A |
| Battery | Skyholic Li-polymer battery | Nominal 22.2 V, 4.5 Ah |
| Battery Charging Boost Converter | Input/Output Range | Input: 3–15 V/~3.5 A<br>Output: 16.2–25.2 V/1 A |
|  | Inductor | 30 μH |
|  | Input/Output Capacitor | 100 μF/220 μF |
|  | Switching Frequency | 90 kHz |

Figure 17 shows the battery charging control result through the MPPT operation when the laser power was 2.874 W/cm$^2$. The converter operated in the MPPT mode because the battery voltage has not reached the 24.5 V set as the full-charge voltage. The sampling period of the MPPT algorithm was set to 0.1 s to reduce the simulation time, and it was observed that the input voltage set point was generate to track the 7.5 V point in the MPP. Since the converter was well controlled by the voltage command, it can be confirmed that the battery was charged with the maximum power generated from the laser PV module.

*3.2. Experimental Results*

Figure 18 shows the experimental configuration of a battery charging system using a laser beam. The laser light source was configured to be generated by passing a multilens array at the rear end of the MFSC-200 device with a wavelength of 1080 nm. The power generated from the laser PV module was supplied to the battery after passing through the prototype boost converter, and the electronic load was connected in parallel to the battery for the load test.

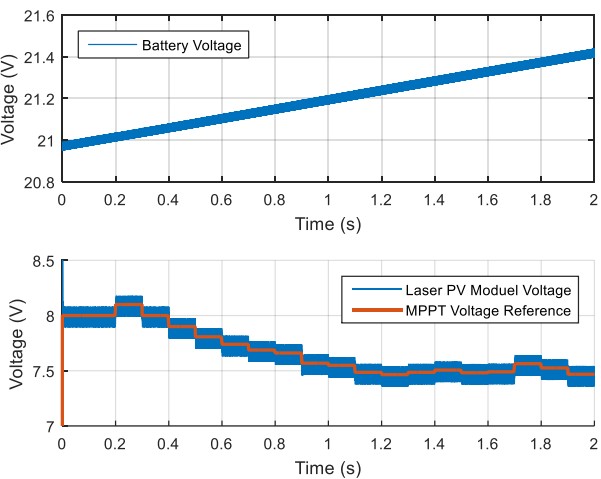

**Figure 17.** Simulation result of charging battery to the maximum power point.

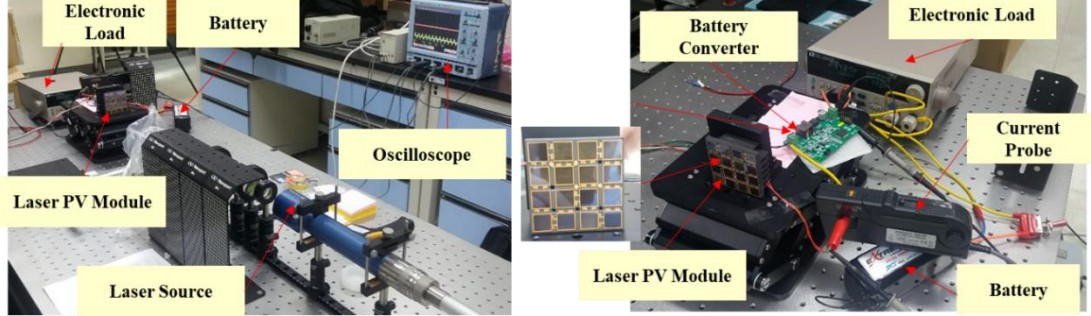

**Figure 18.** Experimental setup for laser wireless power transmission.

Figure 19 shows the main configuration of the prototype boost converter system, where a TMS320f28069 MCU was applied to communicate with the UAV's host controller, perform MPPT functions, and control the converter. Table 2 shows the experimental configuration of the laser wireless power transmission system and the main specifications of the manufactured converter.

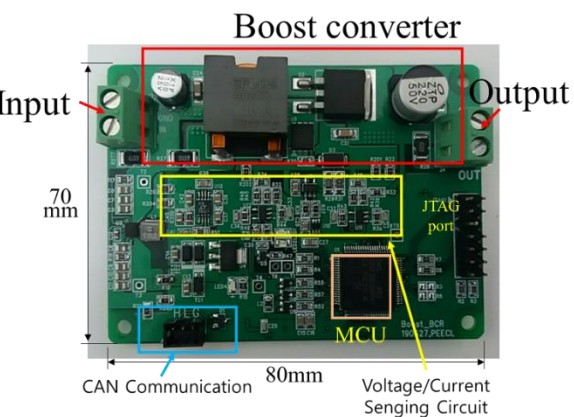

**Figure 19.** Prototype boost converter configuration.

The startup sequence of the battery charging device and the results of the MPPT control test are shown in Figures 20 and 21. The battery was first connected to the converter ("Battery ON") before the laser beam was applied to the converter; subsequently, the laser light source was irradiated to the PV module, and the battery charging device was operated. The activated converter performed a self-diagnostic verification and operated after 5 s ("BCR On") when no faults were encountered.

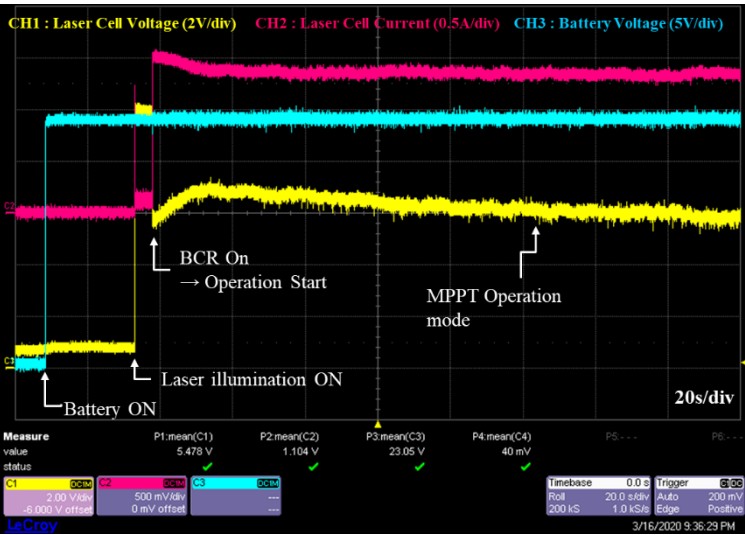

**Figure 20.** Experimental result of maximum power point tracking (MPPT) control of laser PV module for battery charging.

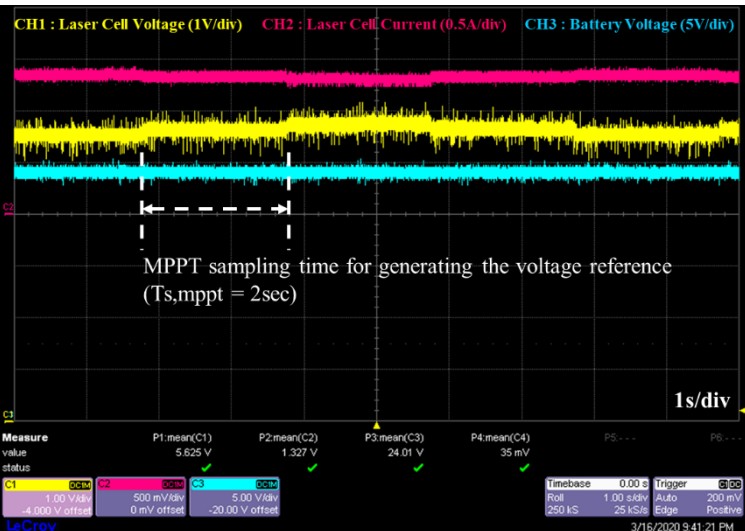

**Figure 21.** Enlarged waveform of MPPT control operation.

In this experiment, because the battery did not reach the full-charge voltage set at 24.5 V, it was operated in the MPPT mode; furthermore, it was confirmed that the battery was charging while moving to the maximum power point of the PV module. Figure 21 shows the enlarged waveform of the converter performing the MPPT operation. The voltage of the laser PV module, indicated by the yellow CH1 waveform, was tracking the maximum power point by changing the operating point at every MPPT sampling period with a 2 s period.

## 4. Conclusions

Herein, a controller design method that reflects the small-signal voltage and current characteristics of a laser PV module for a wireless power system using a laser beam was presented. The laser PV module was fabricated to generate the maximum energy from a laser light source of a specific wavelength (1080 nm in the case of the module used in this study). From the PV module experiment, it was confirmed that the voltage and current characteristics were similar to those of the solar cell module. First, the power generated by the laser PV module varies according to the power of the laser beam. Therefore, to control the operating point to the maximum power generation point that varies

according to the power of the laser beam, the power conversion unit must have an MPPT control function. Secondly, the laser PV module has a characteristic of a small-signal resistance having a negative value similar to that of a solar cell module, and the small-signal resistance value increases as the operating point go to the current source region. Therefore, when designing the controller for controlling the operating point of the laser PV module, the small-signal resistance characteristics of the laser PV module must be reflected. Third, although not described in this paper, the temperature of the PV module increases when a laser beam is irradiated with the laser PV module. As the temperature of the laser PV module increases, the open-circuit voltage of the laser PV module tends to decrease rather than the magnitude of the short circuit current. Therefore, electrical characteristics analysis and control studies according to the temperature rise of the laser PV module are required as future research.

Accordingly, in this paper, a controller design method that can stably control the input voltage in the MPPT mode by inducing the transfer function of the boost converter, reflecting the small-signal characteristics of the laser PV module, was systematically presented. The method proposed herein was verified through simulation results based on Matlab/Simulink and an experiment involving a 25-W-class prototype boost converter.

**Author Contributions:** S.L. contributed to the main idea of this study and wrote the paper. S.L., N.L., and W.C. performed the experiments. J.B., Y.L., and J.P. revised the paper comprehensively. All authors have read and agreed to the published version of the manuscript.

**Funding:** This paper was supported by a Korea Institute for Advancement of Technology (KIAT) grant funded by the Korea Government (MOTIE) (P0002092, The Competency Development Program for Industry Specialist). This work was supported the Korea Institute of Energy Technology Evaluation and Planning (KETEP) grant funded by the Korea government (MOTIE) (20182410105280).

**Conflicts of Interest:** The authors declare no conflict of interest.

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
