# Peer review of "Study on Battery Charging Converter for MPPT Control of Laser Wireless Power Transmission System"

_electronics, doi:10.3390/electronics9101745_

Round 1
Reviewer 1 Report
Some additional comments are as follows:
- What is the main contribution of the paper? The control method for boost converter?? Bold the novelty in the paper.
- In the abstract and conclusion, it is better to explain in a short sentence the advantages of the proposed method.
- The introduction is not complete and the references are not mostly updated. It is suggested to add more related references.
- The second paragraph of the introduction (what you are going to present in the paper) is suggested to move at the end of the introduction.
- The motivation for the proposed method is not clear. It is suggested to mention the disadvantages of other methods and the motivation for the proposed method at the end of the introduction part.
- There are some grammatical errors in the text. It is a good idea to have the paper proofread by a native English speaker.
- The equations 2,3,4,5 are not one equation. It is better to put them in brace if you want to dedicate them to one equation number.
- How did you programed your DSP? Do you have an interface board?
- How did you communicate TMS micro with UAV’s host controller?
- Give some explanation about the electronic load you have used.
Author Response
Thank you for your valuable comment on this paper.
We did our best within the deadline to supplement the contents as shown in the attached file.

Reviewer 2 Report
Journal: Electronics (ISSN 2079-9292) Manuscript ID: electronics-960238 Title: Study on Battery Charging Converter for MPPT Control of Laser Wireless Power Transmission System In this paper, the authors proposed a method for designing the controller of a power converter capable of charging a battery while generating maximum power from a PV module. First, an experiment is conducted to obtain the characteristic data of the voltage and current based on the laser output power of the laser PV module, which generates the maximum power from the laser beam at a wavelength of 1080 nm; subsequently, the small-signal voltage and current characteristics of the laser PV module are analyzed. Because the laser PV module corresponds to the input source of the boost converter used as the power conversion unit, the small-signal transfer function of the boost converter, including the PV module, is derived for the controller design. Therefore, by designing a controller that can stably control the voltage of the PV module in the current source, maximum power point, and voltage source regions defined according to the output characteristics of the laser PV module, the maximum power is generated from the PV module. Herein, a systematic controller design method for a boost converter for laser wireless power transmission is presented, and the proposed method is validated based on simulation and experimental results of a 25-W-class boost converter based on microcontroller unit control. The research is of great interest. The following manuscript has some weakness and here are the most important topics/questions to be dealt with: 1/ Re-write the abstract by providing detailed findings from the experimental studies. 2/ Introduction The Introduction should consist of five paragraphs answering the following five questions: What is the problem? Why is it interesting and important? Why is it hard? Why hasn't it been solved before? (Or, what's wrong with previously proposed solutions?) What are the key components of my approach and results? As above-mentioned questions that should be replied, it will be better that write more relation to the benefits & disadvantages of the blending techniques and more investigate about limitation of previous studies. Also, this part needs more explanations to state clearly objectives & hypothesis of this study at the end of the Introduction part. It should be mentioned to the factors that be shed light by this study. 3/ The literature review is not sufficient with missing some important papers published recently. 4/ Reviewer did not find the reply for the following questions: (i) What value does the paper add? (ii) What is the purpose of the paper? It is not clear, what new knowledge will we gain after reading this work. Corresponding statements should be added into the manuscript in regards to the achievements in this research field. Corresponding statements should be added into the manuscript in regards to the achievements in this research field. Moreover, Authors should exhibit the gap in knowledge which the present study fills. 5/ Figure 3 and figure 4, the numbers in the figures cannot be identified, which needs to be significantly improved. 6/ Figure 15, could the authors provide a detailed error analysis for the data? 7/ Conclusion section is extremal short. The conclusions are very weak and ít requires a deeper analysis of the results. The reviewer suggests carefully read the whole manuscript again before resubmitting to the journal Electronics. Authors should consider above-mentioned remarks in order to revise the manuscript. The reviewer thinks that a publication of the draft manuscript may be possible after a minor revisionAuthor Response
Thank you for your valuable comment on this paper.
We did our best within the deadline to supplement the contents as shown in the attached file.

Round 2
Reviewer 1 Report
I would like to thank the authors for their effort in developing and considering comments. They have addressed all of my comments. I think the paper has good quality now.
I have one other comment as:
1- On page 12, line 265, Table 2 should be changed to Table 1.